# Cross-Sectional Study of the Prevalence of Cobalamin Deficiency and Vitamin B12 Supplementation Habits among Vegetarian and Vegan Children in the Czech Republic

**DOI:** 10.3390/nu14030535

**Published:** 2022-01-26

**Authors:** Martin Světnička, Anat Sigal, Eliška Selinger, Marina Heniková, Eva El-Lababidi, Jan Gojda

**Affiliations:** 1Centre for Research on Diabetes Metabolism, Nutrition of Third Faculty of Medicine, Charles University, 10000 Prague, Czech Republic; anats994@gmail.com (A.S.); eliska.selingerova@lf3.cuni.cz (E.S.); marina.henikova@lf3.cuni.cz (M.H.); eva.ellababidi@fnkv.cz (E.E.-L.); jan.gojda@lf3.cuni.cz (J.G.); 2Department of Internal Medicine, Third Faculty of Medicine, University Hospital Královské Vinohrady, Charles University, 10000 Prague, Czech Republic; 3Department of Pediatrics, Third Faculty of Medicine, University Hospital Královské Vinohrady, Charles University, 10000 Prague, Czech Republic; 4Centre for Public Health Promotion, The National Institute of Public Health, 10000 Prague, Czech Republic

**Keywords:** vegetarian, vegan, children, vitamin B12, B12 supplementation, deficiency

## Abstract

Vegetarian (VG) and vegan (VN) diets in childhood are of growing interest due to their perceived health and environmental benefits. Concerns remain due to the possible disruption of healthy growth and development of children because of the scarcity of evidence-based studies. Among the nutrients of special concern is vitamin B12. Therefore, the Czech Vegan Children Study (CAROTS) decided to examine the relationship between B12 metabolism parameters and B12 intake through diet and supplementation. We analyzed laboratory parameters within *n* = 79 VG, *n* = 69 VN, and *n* = 52 omnivores (OM) children (0–18 years old). There were no significant differences in levels of holotranscobalamin (aB12), folate, homocysteine (hcys), or mean corpuscular volume. However, there was a significant difference in levels of cyanocobalamin (B12) (*p* = 0.018), even though we identified only *n* = 1 VG and *n* = 2 VN children as B12 deficient. On the other hand, we identified *n* = 35 VG, *n* = 28 VN, and *n* = 9 OM children with vitamin B12 hypervitaminosis (*p* = 0.004). This finding was related to a high prevalence of over-supplementation in the group (mean dose for VG 178.19 ± 238.5 µg per day; VN 278.35 ± 394.63 µg per day). Additionally, we found a significant (*p* < 0.05) difference between B12, aB12, and hcys levels of supplemented vs. non-supplemented VG/VN children. This can show that the intake of vitamin B12 via diet in the VG group might not be sufficient. Secondly, we analyzed a relation between supplement use in pregnancy and breastfeeding and its impact on vitamin B12 levels of children aged 0–3 years. Out of *n* = 46 mothers, only *n* = 3 (e.g., 6.5%) were not supplemented at all. We have not identified any clinical manifestation of B12 deficiency and only *n* = 1 child with low serum cobalamin, a child who did not receive vitamin B12 supplementation and whose mother took only low doses of vitamin B12 (25/µg/day).To conclude, we did not observe any life-threatening or severe consequences of laboratory-stated vitamin B12 deficiency; thus, our group was well supplemented. On the other hand, we have identified many subjects with vitamin B12 hypervitaminosis of unknown impact on their health. Further research and new guidelines for B12 supplementation among VG and VN children are needed.

## 1. Introduction

The popularity of plant-based diets has increased over the past decades, mainly as a consequence of perceived health and environmental benefits and ethical reasons [1,2]. There are several patterns with variable restriction of animal products: vegan diet (VN), which excludes all animal products, e.g., meat, fish, dairy, eggs, and honey, and vegetarian diet (VG), which can be divided into lacto-ovo-vegetarian (excluding only meat and fish), lactovegetarian (excluding meat, fish, and eggs) and ovo-vegetarian (excluding meat, fish and dairy) [3]. Reducing animal sources of food at the population level brings potential concerns about the long-term safety of the restrictive diets, and health care authorities should be aware of how to tackle this transition in terms of managing potential risks [1].

Nevertheless, data regarding B12 status among pregnant and lactating vegan women, infants, and children is very limited. There are highly diverse cross-sectional studies on VG or VN diets during childhood, which are of small sample size and are often outdated (mainly from 1970–1990) [4,5]. Since then, there has been an increase in the popularity of plant-based diets and a change in the food market, which is now offering a variety of plant-based milk alternatives, as well as meat substitutes. These have not been available before and are often fortified with B12 [6]. Furthermore, our market provides several special supplements for the VN and VG population. Although much information is available online, dietary guidelines on B12 intake in children on plant-based diets are scarce. On the other hand, the percentage of VN and VG mothers is increasing, as well as the risk of vitamin B12 deficiency in this particularly sensitive period of the child’s development [2].

### Vitamin B12

Vitamin B12, cobalamin, cannot be synthesized within the human body, making it an essential nutrient. Instead, B12 is synthesized in soil and gut bacteria and accumulates in animal tissues [7]. Even though vitamin B12 is a water-soluble vitamin, its largest reserves are found in liver tissue cells and animal-based food is a major source of the nutrient for humans [8]. Syndromes associated with B12 deficiency are macrocytic anemia, which leads to failure to thrive and delayed growth in children [9]; polyneuropathy or retinopathy with potentially long-lasting consequences. To prevent deficiency, vitamin B12 has to be acquired from the diet or via supplements. It is known that, similar to vegans, vegetarians who do not consume enough vitamin B12 supplementation have a higher risk of deficiency than people eating a regular diet [10]. With the increase in popularity of plant-based diets worldwide, it is essential to consider the possible adverse effects of vitamin B12 deficiency [1,2]. Unfortunately, vitamin B12 supplements are not always well-received by the vegan community. A systemic review from 2014 shows alarming data that the prevalence rates of cobalamin deficiency in vegetarian and vegan adults and the elderly range from 0% to 86.5%, with much higher numbers among vegans [10]. In one original research, more than 78% of Slovak vegans have got laboratory signs of cobalamin deficiency [11]. A balanced maternal diet during pregnancy and breastfeeding is critical for the mother’s well-being and her offspring’s growth and development [2]. Inadequate maternal vitamin B12 consumption during breastfeeding can result in low vitamin B12 levels in breast milk, which can cause lifelong developmental deficits in their children [2]. The incidence of cobalamin deficiency in breastfed babies is primarily determined by the mother’s vitamin B12 status; therefore, vegetarian and vegan populations are particularly at risk [12,13,14]. Since folic acid intake from a plant-based diet can mask a vitamin B12 deficiency, we assumed that levels of total vitamin B12 and MCV alone are not sufficient to determine the metabolism of vitamin B12 [15,16,17]. Today, the most reliable parameter determining the amount of vitamin B12 in the human body is the level of methylmalonic acid, which, unfortunately, is not routinely available in every laboratory, especially in the Czech Republic. Therefore, we can use another parameter, active vitamin B12 (holotranscobalamin), which meets the requirements of vitamin B12 metabolism in the human body and is available for measurement [17]. Besides, it is recommended to evaluate the level of homocysteine which is also elevated in case of vitamin B12 deficiency [18]. The aims of our study are (1) to analyze laboratory parameters of vitamin B12 metabolism in the VG/VN group and to compare them with OM children; (2) to analyze the difference within the VG/VN group regarding supplement use (e.g., supplemented and not-supplemented) and discern whether a sole VG diet is a sufficient dietary source of vitamin B12; and (3) to see whether B12 supplementation habits in mothers during pregnancy and breastfeeding impact offspring’s B12 laboratory status.

## 2. Materials and Methods

### 2.1. Study Design and Participants

Due to the unavailability of more precise data for sample size calculation, the prevalence of cobalamin deficiency was reported for Czech adult vegans [19] and previously described in review for the paediatric population [10], together with the reported prevalence for the omnivorous population [20] were used for estimation of the expected differences between groups. Based on these assumptions, together with realistic expectations regarding the possible recruitment in the Czech Republic, a sample size of at least 165 children (power = 80%) was established as a target.

Finally, a sample of 203 children, consisting of 80 VG, 71 VN, and 52 OM controls, was recruited for cross-sectional analysis. Children were enrolled and examined between November 2019 and July 2021. The recruitment of the volunteers was performed through collaborating general practitioners and other specialists, advertisements over social media, and websites focused on veganism. At the end of the recruitment period *n* = 1 VG children and *n* = 2 VN children were excluded because their parents did not consent to the venepuncture. Hence the final sample consisted of *n* = 79 VG, *n* = 69 VN and *n* = 52 OM children. Examination of the children took place at the Department of Paediatrics, Third Faculty of Medicine, Charles University, University Hospital Královské Vinohrady. The inclusion criteria for participants were (1) self/parent-reported vegetarian or vegan or omnivorous children, (2) age of 0–18 years. Subjects with any chronic disease, namely disease that could affect nutrient absorption (e.g., enteropathy, pancreatic insufficiency, metabolic diseases such as phenylketonuria, or fructose malabsorption criteria) were not enrolled—see Figure 1. A participant was self-classified as “Omnivore” (OM) if eating meat, dairy, and eggs, as “Vegetarian” (VG) if not eating meat/meat products/fish, but consuming dairy/eggs, and as “Vegan” (VN) if not consuming any food of animal origin (except for honey). Based on ex-post dietary data if the participant claimed to eat dairy/eggs more than once per week in an obtained dietary record, they were reclassified as a vegetarian. If the participant claimed to eat meat, meat products, or fish more than once per week, they were reclassified as omnivore.

From our total number of *n* = 148 participants, we selected a subgroup of *n* = 46 children, according to the following criteria: (a) vegetarian or vegan children, (2) age of 0–3 years, (3) children of vegan or vegetarian mothers, (4) breastfed at the time of examination, to analyze the relationship between B12 intake in vegan and vegetarian mothers during the period of pregnancy and breastfeeding and its influence on their child’s growth and development. See Figure 1. The study was conducted according to the Declaration of Helsinki guidelines and approved by the Ethics Committee of the Third faculty, Charles University, Prague, and Ethics Committee of Faculty Hospital Královské Vinohrady. All examinations were performed with parental written consent, with no financial incentives.

### 2.2. Personal History and Examination

We took a personal history (from mothers or adolescents), including self-identification with one of the VN/VG groups. Specific focus was placed on information about vitamin B12 supplementation (physician-supervised questionnaire): its chemical form (e.g., adenosylcobalamin, cyanocobalamin, methylcobalamin, hydroxycobalamin), frequency of usage (e.g., daily, twice, or three times a week), the regularity of use (e.g., regular, not regular), in which form it was consumed (e.g., pills, capsules, drops, sublingual powder), and the dosage (in µg). Furthermore, we asked the mothers for information regarding their diet habits, breastfeeding habits (e.g., fully, partially), supplements intake during lactation, and vitamin B12 supplementation (same modalities as for the children) during pregnancy lactation.

### 2.3. Biochemical Analysis

The biomarkers of interest were serum holotranscobalamin (active B12), cyanocobalamin, folate, homocysteine, mean corpuscular volume (MCV), and haemoglobin, which we analyzed from participants’ blood. Peripheral blood samples were obtained in most cases after an overnight fast. However, in some cases (nurslings, infants), blood samples could be obtained only after a breastfeeding or a small and light breakfast. Serum analyzes were performed immediately in the ISO-certified institutional laboratory by validated, routine methods. Both cobalamin and folate chemiluminescence immunoassay were analyzed automatically on a Cobas 8000 system (Roche Diagnostics GmbH, Mannheim, Germany). For holotranscobalamin electrochemiluminescent Elecsys active B12 assay, analyzed automatically on an Abbott Architect i200SR. Homocysteine was assessed using an Advia Centaur XP (Siemens Healthcare Diagnostics, Tarrytown, NY, USA). The haemoglobin (HGB) in the whole blood was measured using photometry, the haematocrit using direct impedance measurement, and the MCV was calculated as haematocrit/red blood cell count (RBC). For cobalamin, holotranscobalamin, folate, and homocysteine reference values provided by the manufacturer were used.

### 2.4. Nutritional Assessment

Dietary intake was assessed using 3-day weighed dietary records in our VG participants. Using electronic kitchen scales, the parents weighed and recorded all foods and beverages consumed by the participating children over three days (weekdays and weekends). The participating families chose the day of the beginning of dietary recording within a given period. When exact weighing was not possible—e.g., in case of eating out—household measures (e.g., spoons, cups) and a photo booklet with foods in children’s portion sizes, supplemented with special VG and VN foods, allowed semi-quantitative recording. The study staff assessed missing data, requesting the information from the parents via e-mail. Breast milk intakes were estimated from mothers’ registrations and general recommendations for breast milk intake. Energy and nutrient intakes were calculated using the food composition database Nutriservis PROFI (Forsapi Ltd., Praha, Czech Republic). The energy and nutrient contents of commercial food products, i.e., processed foods and ready-to-eat meals or snack foods, were estimated by recipe simulation using labelled ingredients and nutrient contents. Nutriservis PROFI is continuously updated by adding those products or supplements recorded by dietitians. In cases where some foods from the dietary record were missing in the Nutriservis PROFI database, they were manually entered into the database by a dietitian. When entering food data into the database, the dietitian copied the values of calories, macronutrients, and micronutrients from product packaging. For products whose nutrition facts labels did not state the amount of B12 on the package, the dietitian used the USDA database (https://fdc.nal.usda.gov/index.html, accessed on 21 May 2021), the Fitbit application database (https://www.fitbit.com/foods, accessed on 21 May 2021), in rare cases, the Eatthismuch database (https://www.eatthismuch.com/, accessed on 21 May 2021). If the products were not found in one of the databases mentioned above, the dietitian used the composition on the package and manually calculated the amount of B12. The dietitian then entered all the obtained data into the Nutriservis PROFI database (see Table 1).

### 2.5. Anthropometrics

Bodyweight (in kilograms) and body height (in centimeters) were measured as the average of three independent measurements using calibrated scales. Nurslings’ and toddlers’ length was measured with an infantometer, older children’s height was measured with a stadiometer. Consequently, we entered the values were transformed to percentile values using a standard percentile graph validated for use in the Czech Republic (publicly accessible online: “6. National Anthropological Research 2001”) [22].

### 2.6. Data Analysis

In the case of continuous variables (anthropometric measurements, biochemical markers) with normal distribution, mean values together with their standard deviations are shown and statistical significance is assessed using parametric analysis of variance. In variables where normality test (Shapiro–Wilk normality test) showed a possible disruption of the normality assumption, median, and interquartile range (IQR) are reported and non-parametric test (Kruskal–Wallis test) is used to assess the statistical significance of the difference. For categorical variables, the absolute number of children in a category, together with proportions, are reported. The statistical significance of the difference in proportions is evaluated using Fisher’s exact test. Multivariable logistic regression models with the odds ratios (95% confidence intervals) of hypovitaminosis as well as hypervitaminosis defined based on assessed laboratory markers were carried out to describe the differences in risk between the dietary groups. Logistic regression models were adjusted for age and sex. Analysis of biochemical markers related to cobalamin deficiency was done separately for children of 3 years of age and younger (“Infants and toddlers”) and children older than 3 years (“Schoolers”). The vitamin B12 deficiency was stated as levels of aB12 or B12 under a lower reference limit (for more details please see Table 2).

Besides the general analysis, subgroup analysis of children breastfed at the time of examination was carried out for VN and VG children, describing the supplementation habits of the children, the supplementation of the mother during pregnancy, as well as lactation, and the differences in relevant biochemical markers concerning the supplementation of children. All analyzes were performed using R (R: A language and environment for statistical computing) (R Core Team, Foundation for Statistical Computing, Vienna, Austria, 2018; URL, https://www.r-project.org/, accessed on 12 December 2021, using the Tidyverse package (Hadley Wickham, 2017). Tidyverse: Easily Install and Load the ‘Tidyverse’. R package version 1.2.1. https://CRAN.R-project.org/package=tidyverse, accessed on 12 December 2021) and ggpubr package (Alboukadel Kassambara (2018) ggpubr: ‘ggplot2’-Based Publication Ready Plots. R package version 0.1.7 https://CRAN.R-project.org/package=ggpubr, accessed on 12 December 2021). A two-sided *p*-value of 0.05 was used to denote statistically significant associations.

## 3. Results

### 3.1. Sample Characteristics

The final sample consisted of the *n* = 200 children, e.g., *n* = 79 VG, *n* = 69 VN and *n* = 52 OM. The mean age of participants was 5.8 years (±4.1) for VG, 4.8 years (±5.8) for VN and 6.7 years (±5.6) for OM and ranged from 0.5 to 18.5 years, maximally. There were significantly more participants under 3 years of age in the VN group than in VG and OM (*p* < 0.001), therefore the median was 4.5 (IQR 2.6–8.0) for VG, 2.0 (IQR 1.0–6.0) for VN and 4.5 (IQR 2.0–10.9) for OM (*p* = 0.003). The sample distribution based on the age of participants is visualized in the histogram (please see Appendix A). There were no significant differences in sex, height percentile, weight percentile, or BMI percentile, but we observed a higher number *n* = 7 of VN children with lower BMI e.g., <3. percentile (*p* = 0.003). For more details, please see Table 3a,b, wherein we split our study group into two subgroups: “Toddlers” e.g., children aged 0.5–3.0 years and “Schoolers” e.g., children aged 4.0–18.5 years—see also Table 4a,b and Table 5a,b. The Histogram showing age distribution could be find in Appendix A.

### 3.2. B12 Dietary and Supplement Intake Analysis

According to the data summarized in Table 6, there was a significant difference in the supplement use (*p* < 0.019), regularity (*p* = 0.014), and forms—chemical (*p* = 0.009) and medical (*p* = 0.048)—between VG and VN. We defined supplement use according to questionnaires we filled with patients during history collection. If a participant did not take any supplements of vitamin B12, they were classified as “No” in their supplement use. If they took any supplements with any content of vitamin B12, they were classified as “Yes”, and if they used any kind of vitamin B12 supplements but stated they could not recall the regularity, they were classified as “Irregular”. Regularity was assessed according to the frequency of use: every day, three times a week, twice a week, and once per week. We also assessed the chemical form, e.g., methylcobalamin, adenosylcobalamin, hydroxymethylcobalamin, and cyanocobalamin. The medical form stated whether such were drops or pills.

Regarding supplement use, *n* = 10 parents of VN children reported that they did not provide any cobalamin supplements compared with *n* = 25 parents of VG children. Irregular cobalamin supplement use was reported by *n* = 2 parents of VN children and *n* = 2 parents of VG children. The median dose of vitamin B12 taken in a single dose was 86.50 ug (IQR 250.0) for VG and 98.6 µg (IQR 210.0) for VN. Mean dosage 178.9 ± 238.50 ug for VG and 278.35 ± 394.63 ug for VN. OM children vitamin B12 supplement use was not assessed because we did not anticipate vitamin B12 deficiency in healthy omnivorous controls. The dietary intake was calculated as an average from the reported 3 days and was 1.9 µg for VG (*p* < 0.001), thus lower than recommended daily intake. In vegans, there is no reliable dietary source of B12; therefore, all vegans in our analysis have reported intake (without supplementation) of 0 µg/day. A detailed plot showing the distribution of intake among vegetarians can be found in Appendix A.

### 3.3. Differences between Dietary Groups in Selected Blood Markers Results

The mean differences in evaluated biomarkers are reported in Table 7. We observed a statistically significant mean difference (*p* = 0.019) in B12 levels between the VN, VG, and OM group, with the omnivore group having the lowest mean value. However, there was no significant difference among the groups in the mean values of the other markers of B12 metabolism. Contrarily, we identified *n* = 35 VG, *n* = 28 VN, and *n* = 9 OM children meeting the laboratory diagnosis of vitamin B12 hypervitaminosis (*p* = 0.004); this phenomenon was more evident in “Schoolers” (for more details see Table 8 and Table 9).

Non-supplementing vegetarians are at the same risk of vitamin B12 deficiency as vegans not using supplements (*p* < 0.001). This is supported by their overall low B12 intake from diet, vitamin B12 supplements excluded. Those data are visualized in Figure 2 and Figure 3.

### 3.4. Subanalysis: B12 Status of Breastfed VG and VN Children Aged 0–3 Years According to B12 Supplementation Habits, Breastfeeding, and Mothers’ Supplementation Habits during Lactation and/or Pregnancy

Our subanalysis consisted of *n* = 46 VG/VN children breastfed at the time of the examination, of whom 12 were VG and 34 were VN. We studied the supplementation habits of the children and their mothers during pregnancy and lactation.

According to the data in Table 10, we found that 78.3% (*n* = 36) of the mothers were supplemented by any form of vitamin B12 during both pregnancy and lactation, 13.0% (*n* = 6) were supplemented during breastfeeding alone, and 2.2% (1) during pregnancy alone. On the other hand, 6.5% (*n* = 3) of the mothers were not supplemented at all; two of them were VN.

Furthermore, we wanted to check the supplementation habits of children in this subpopulation. According to the personal histories we took from mothers, we found out that of *n* = 46 children, *n* = 33 (71.7%) were supplemented by some form of vitamin B12, whereas, among those non-supplemented, were *n* = 6 VG and *n* = 7 VN children; in total 28.3% of our selected subgroup. In addition, we asked the mothers for information regarding their breastfeeding habits and duration: fully breastfeeding, partially breastfeeding, and whether they were breastfeeding at the time of the child’s examination. Children were found to be fully breastfed over an average of 6.65 months and partially breastfed over an average of 16.24 months. There were significant differences in levels of aB12, B12, and hcys between VG/VN children who regularly used supplements with vitamin B12 and between those who did not. For details, see Table 11a,b.

We did not identify any clinical manifestation of B12 deficiency, except in one child, who did not receive B12 supplementation, and whose mother took only a low dose of B12 (25/µg/day) during pregnancy and breastfeeding. On the contrary, *n* = 1 child was not diagnosed with laboratory vitamin B12 deficiency despite the absolute absence of supplementation even in his mother’s diet. However, he consumed a lactoovovegetarian diet, therefore, we assume he had a sufficient natural intake from his diet. *n* = 2 VG children who did not take vitamin B12 supplements were not diagnosed with a laboratory deficiency, but their mothers took high doses of vitamin B12 during breastfeeding. Finally, *n* = 9 non-supplemented children were breastfed by an adequately supplemented mother during pregnancy and breastfeeding, of which *n* = 6 VN.

Based on the results, we assume that the mother’s sufficient supplementation with vitamin B12 during pregnancy and breastfeeding leads to an adequate saturation of the breastfed child with vitamin B12, even in the absence of the child’s supplementation, whereas the lack of supplementation during pregnancy and breastfeeding and the subsequent absence of supplementation of VN children leads to vitamin B12 deficiency.

## 4. Discussion

In our current study involving *n* = 148 Czech VG/VN children and *n* = 52 OM controls, we observed that (1) there is a very low prevalence of vitamin B12 deficiency in the VG/VN group because of (2) high awareness and use of vitamin B12 supplementation. On the other hand, (3) there is a higher risk of developing laboratory vitamin B12 hypervitaminosis within the VG/VN group. Furthermore, we can describe (4) that self-reported VG children often do not consume a sufficient amount of animal products (e.g., dairy and eggs), therefore their recommended daily intake for vitamin B12 is not always met, and they tend to have lower levels of cyanocobalamin when not supplemented as VN children are. (5) Moreover, we identified that vitamin B12 supplementation in vegan mothers is crucial for maintaining physiological levels of vitamin B12 in their children. (6) In addition, we observe a higher number of VN children with BMI percentile < 3.

Despite the existence of several cases in which VG/VN children suffered from severe manifestations of vitamin B12 deficiency [13,23,24] and that some other works describe an insignificant percentage of vitamin B12 deficient VG/VN children [25] or adults [19,26] we identified only *n* = 3 VG/VN children meeting criteria for laboratory vitamin B12 deficiency and none of them had any clinical manifestations. On the other hand, many of our participants had developed a laboratory hypervitaminosis of vitamin B12. This can be due to the lack of supplementation guidelines for the vegan and vegetarian paediatric population in the Czech Republic and the inconsistency of recommendation across the scientific world [27,28,29,30,31], both of which lead to higher and more frequent supplementation to prevent deficiency. The consequence of long-term hypervitaminosis of vitamin B12 has not yet been described in the literature, as there has not been a population using an artificially high dosage of vitamin B12. New epidemiological studies in the adult population describe a higher risk of cancer and cardiovascular risk in people with laboratory hypervitaminosis of vitamin B12. However, causality has not yet been determined [32,33]. Further analysis shows that self-reported VG children often do not meet the recommended daily intake for vitamin B12 via dairy products and eggs. When not supplemented, they tend to have lower serum levels of vitamin B12, which also manifests as higher levels of homocysteine. This finding indicated that Czech VG children are prone to vitamin B12 deficiency without taking any supplements, the same as VN children. This finding is in accordance with similar scientific works [34,35,36]. In addition, we performed an analysis of vitamin B12 deficiencies among the most vulnerable group (e.g., breastfed children aged 0–3 years of VG/VN mothers). Our analysis showed that nearly 28.3% of the children were not supplemented at all, and 6.5% of their mothers did not take any supplements. However, only *n* = 1 VN child was classified as vitamin B12 deficient, showing that the rest were properly saturated through their mother’s regular supplementation during pregnancy and lactation. Therefore we could say that sufficient supplementation with vitamin B12 during pregnancy and lactation, particularly among VN mothers, is crucial for adequate saturation of their children, especially when these children do not receive any form of artificial supplementation. This is supported by other research [37]. In addition, we did not register any case of life-threatening vitamin B12 deficiency in this vulnerable group, which is in contrast with the Czech series of case reports [12].

An unexpected finding, apart from the main hypothesis of our study, is that we observed a higher number of *n* = 7 VN children with BMI < 3rd percentile. Some studies point out that VG/VN children tend to be leaner or smaller than OM children [38,39,40]. These findings also raise the question of whether a purely plant-based diet can provide enough nutrients for proper child development. The other question concerns what the best laboratory method for assessing vitamin B12 deficiency is. Due to the low number of vitamin B12-deficient children, we could not determine which laboratory method is more suitable for the diagnosis of vitamin B12 deficiency. Nevertheless, our data show that changes in MCV are not sensitive enough, and levels of B12 and aB12 seem to be more expedient (in our environment, where levels of MMA are not regularly assessed). This theory is in line with other scientific findings [2,19]. A possible limit to the study is its size. However, from a study already conducted in the Czech adult population in our region, we know that 16.9% of the vegan study population was cobalamin deficient, as well as 5.56% of the supplement users, 17.9% of the irregular supplement users, and 52.9% of the non-users. Compared with non-vegans, Czech adult vegans had OR 7.05 (1.6–31.06) for being classified as deficient [19]. The risk for deficiency was strongly associated with lower supplement use among adults, which can explain why such difference was not replicated in the paediatric population, with its higher compliance to supplement use. It should also be noted that, due to the limited access to vegan communities, current conclusions and guidelines are often based on studies with an even lower number of participants. Another limit of the study is possible selection bias. The recruitment of individuals took place through general practitioners and social networks (moreso for mothers of vegans and vegetarians). As evaluated lately, these websites are not visited by adolescents; therefore, the number of children examined at adolescent age is very low. Furthermore, patients approached in this way are more motivated to cooperate with doctors, are more aware of possible deficiencies in the diet, and are probably less participatory in a so-called alternative lifestyle. Observational bias could also influence the outcome of the study, where a three-day record of eating habits was completed purely individually by parents without external control. Parents tend to have information that better suits the knowledge of a healthy and balanced plant-based diet and can feed children better and healthier on the recorded days. Furthermore, there is a general issue with the accuracy of the information collected concerning the weight of served and eaten food. However, despite everything, this method provides more detailed information than FFQ (food frequency questionnaire). We used weighted dietary records because they provide the best estimate for children [41]. The prospective dietary record did not depend on the parents’ ability to recall their children’s food intake. Parents were instructed to maintain their usual diets. Missing information was immediately collected from parents. Our study did not include pregnant females or lactating females, who may be particularly vulnerable to the consequences of cobalamin deficiency. Instead, we assessed only supplementation habits and laboratory findings in their offspring. Indeed, borderline cobalamin status in pregnant women can be associated with an increased risk of acquired newborn B12 deficiency, and cobalamin alone may not be sufficient to diagnose deficiency among them, which is why MMA tests are required [42]. Last but not least, we are aware that there are more reliable markers of cobalamin deficiency, namely methylmalonic acid (e.g., MMA), which can be used; therefore, further analysis of our group of patients using levels of MMA would be certainly needed and widely beneficial [15,43,44]. The main strength of the study is the relative balance between the study groups, with only a slightly higher number of children aged 0–3 years in the VN group and no significant differences in sex, height, weight, or BMI. In addition, there is extensive data about supplement use; its regularity, dosage, chemical, and medical form.

The purpose of this study was to describe vitamin B12 metabolism and supplement use among Czech VG/VN children aged 0–18 years and to discern whether they are at particular risk of vitamin B12 hypovitaminosis. We did not observe any severe cases of vitamin B12-deficient children and only *n* = 3 children in the VG/VN group have been diagnosed with laboratory vitamin B12 deficiency. On the other hand, the awareness of the need for artificial supplementation was high in our participants and more often led to the laboratory diagnosis of vitamin B12 hypervitaminosis of unknown impact. Unfortunately, despite the high level of awareness in our VG/VN group, several mothers did not take any form of vitamin B12 during pregnancy or lactation, due to a lack of information regarding supplementation. This puts their children at risk of developing a hypovitaminosis of vitamin B12 with all the attendant consequences. Our study provides new and exclusive data about vitamin B12 saturation and supplementation habits among Czech VG/VN children. According to our knowledge, a study of this scope has never been carried out on VG/VN children in the Czech Republic. Furthermore, it brings a new light on the general health, behaviour (e.g., supplement use), and nutrient intake of our VG/VN children. Considering the increasing prevalence of plant-based diets, especially in the paediatric population, there is an urgent need for proper guidelines accessible for health professionals for assessing cobalamin deficiency using suitable blood markers. In our environment, we must consider that our general practitioners usually rely on the assessment of cyanocobalamin alone, or along with MCV, which does not reflect true cobalamin deficiency among vegan and vegetarian children. Therefore, we think it is wise to recommend analyzing holotranscobalamin levels as a primary screening for vitamin B12 deficiency in VG/VN children. As for now, there are no sufficient guidelines for vitamin B12 supplementation in our country, and VG children are not considered a population at risk of vitamin B12 deficiency. Unfortunately, our data analysis suggests that not-supplemented VG children are at risk of vitamin B12 deficiency, therefore we recommend changing the supplementation approach and making specific and accessible guidelines. Due to the high prevalence of vitamin B12 hypervitaminosis in our study group, there is a need to establish a safe and specific dosage of vitamin B12 for the VG/VN paediatric population. In addition, we recommend administering a sufficient dose of an artificial vitamin B12 supplementation to a child of a VN mother, who does not take any supplements during lactation. This is supported by other studies [45]. Our research may potentially stimulate future research to determine the exact adequate and safe dosage of vitamin B12 supplementation for VG/VN children. In addition, it encourages an evidence-based discussion concerning proper medical guidelines and recommendations focusing on the paediatric population and plant-based diets in eastern European countries.

## 5. Conclusions

We conclude that Czech VG/VN children may be at the risk of cobalamin (e.g., holotranscobalamin) deficiency when not adequately supplemented. On the other hand, we did not observe any life-threatening or severe consequences of laboratory-stated vitamin B12 deficiency, as mentioned in some case reports across the world. In addition, we found many cases of vitamin B12 hypervitaminosis of unknown impact on children’s health and development. We observed that self-reported VG subjects often did not consume a sufficient amount of animal products, e.g., dairy and eggs in the diet, thus cannot meet the recommended daily intake of vitamin B12 in its natural form. Furthermore, vegetarian children are also prone to be vitamin B12 deficient when not adequately supplemented. In addition, we concluded that artificial vitamin B12 supplementation is needed in VG/VN mothers, especially when their offspring are also VG/VN and do not use any vitamin B12 supplements. It also might be noted that we observed a higher number *n* = 7 of VN children with lower BMI e.g., <3. percentile. Our conclusions go hand in hand with other recent findings that support our hypothesis, except for a significant number of VG/VN children with laboratory vitamin B12 hypervitaminosis [25,35].

## Figures and Tables

**Figure 1 nutrients-14-00535-f001:**
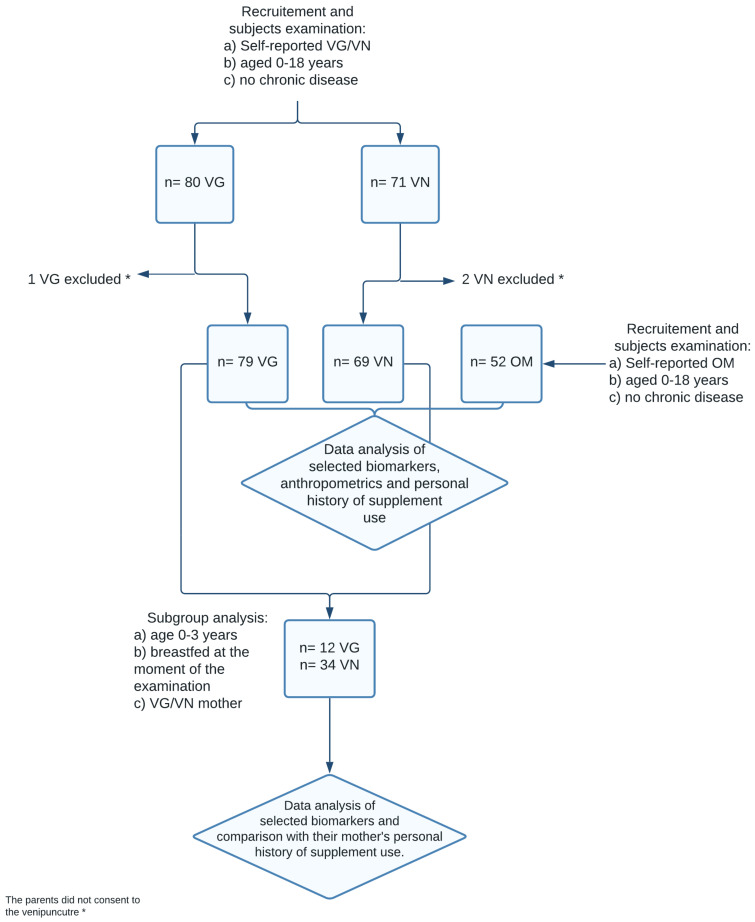
Flow chart of the study design.

**Figure 2 nutrients-14-00535-f002:**
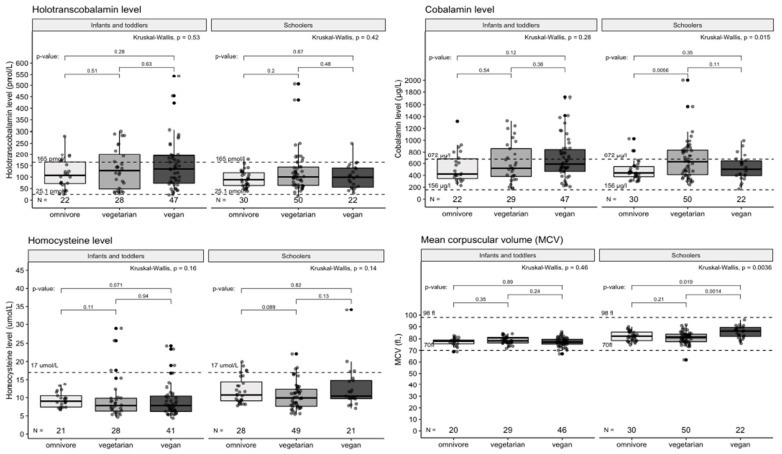
Laboratory markers of vitamin B12 metabolism in diet groups. Cross-sectional comparison of 69 VN, 79 VG and 52 OM. MCV (fl) = mean corpuscular volume. Dashed lines indicated the reference interval of the marker.

**Figure 3 nutrients-14-00535-f003:**
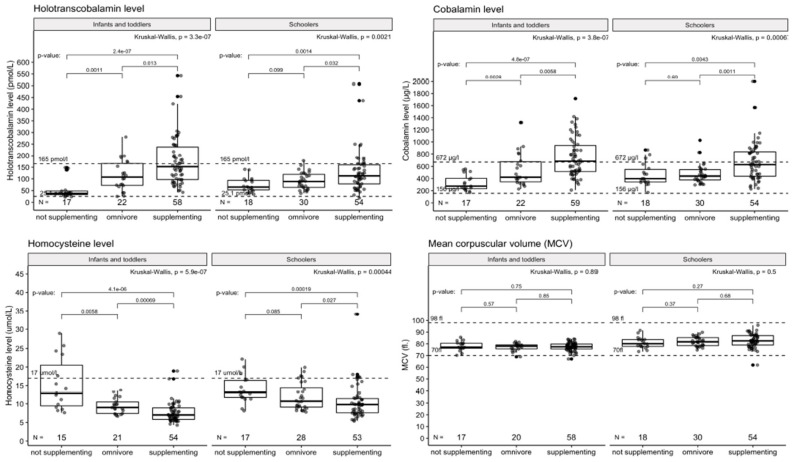
Laboratory markers of vitamin B12 metabolism according to supplement use in the VG/VN group. Cross-sectional comparison of 69 VN, 79 VG, and 52 OM, divided by age group. MCV (fl) = mean corpuscular volume. Dashed lines indicated the reference interval of the marker.

**Table 1 nutrients-14-00535-t001:** Recommended daily intake of vitamin B12 according to age groups.

Group	Age	B12 [μg]
nursling	0–12 months	0.4
children	1–3 years	0.5
	4–6 years	0.8
	7–10 years	1.0
	11–14 years	1.2
adolescents	13–18 years	1.5

According to: Gandy, J., Madden, A., and Holdsworth, M. (2020). Oxford Handbook of Nutrition and Dietetics. Oxford University Press [21].

**Table 2 nutrients-14-00535-t002:** Reference intervals of the laboratory parameters.

	Lower Reference Limit	Upper Reference Limit
**holotranscobalamin**	25.1 pmol/L	165 pmol/L
**cyanocobalamin**	156 µg/L	672 µg/L
**folate**	5.38 µg/L	40 µg/L
**homocystein**	3.0 µmol/L	17.0 µmol/L
**hemoglobin**		
6 months–2 years	105 g/L	135 g/L
2 years–6 years	115 g/L	135 g/L
6 years–12 years	115 g/L	155 g/L
12 years–15 years female	120 g/L	160 g/L
12 years–15 years male	130 g/L	160 g/L
**mean corpuscular volume**		
6 months–2 years	70 fl	86 fl
2 years–6 years	75 fl	87 fl
6 years–12 years	77 fl	95 fl
12 years–15 years female	78 fl	102 fl
12 years–15 years male	78 fl	98 fl
15 years + Female and male	82 fl	98 fl

**Table 3 nutrients-14-00535-t003:** (**a**) Description of the basic characteristics of VG/VN/OM group. (**b**) Description of the basic characteristics of VG/VN/OM group.

(**a**)								
**Group**	** *n* **	**Sex**		**Median Age**	**IQR**	***p*-Value**	**Min Age**	**Max Age**
		**Female**	**Male**					
VN	69	31	38	2.0	1.0–6.0	0.003	0.5	18
VG	79	44	35	4.5	2.6–8.0		0.5	18
OM	52	25	27	4.5	2.0–10.9		0.75	18.5
(**b**)								
**Variable**		**Omnivore**		**Vegetarian**		**Vegan**		***p*-Value**
		**Med**	**IQR**	**Med**	**IQR**	**Med**	**IQR**	
height percentile		45.0	23.0–70.0	48.0	24.7–70.0	48.0	28.0–73.0	0.883
weight percentile		44.0	16.0–65.5	47.0	20.5–70.0	40.0	11.0–65.0	0.386
BMI percentile		40.0	19.5–55:0	42.0	25.0–67.0	35.0	14.0–65.0	0.529
		***n* (total)**		***n* (total)**		***n* (total)**		***p*-Value**
height ≤3 perc.		2 (52)		3 (69)		2 (79)		1.000
weight ≤3 perc.		5 (52)		4 (69)		6 (79)		0.588
BMI ≤3 perc.		1 (52)		0 (69)		7 (79)		0.003

*n* = of children in a given percentile category, total = total number of children in the diet group, the *p*-value was calculated using Kruskal–Wallis test for continuous variables and Fisher’s exact test for categorical variables. *p* < 0.05 was considered significant. med = median. perc. = percentile.

**Table 4 nutrients-14-00535-t004:** (**a**) Description of the basic characteristics of VG/VN/OM “Infants and Toddlers” (aged 0–3 years); (**b**) description of the basic characteristics of VG/VN/OM “Infants and Toddlers” (aged 0–3 years).

(**a**)								
**Group**	** *n* **	**Sex**		**Median Age**		**IQR**		***p*-Value Age**
		**Female**	**Male**					
VN	47	19	28	1.5		0.8–2.0		0.076
VG	29	17	12	2.0		1.5–2.8		
OM	22	8	14	1.8		1.1–2.2		
(**b**)								
**Variable**		**Omnivore**		**Vegetarian**		**Vegan**		***p*-Value**
		**Med**	**IQR**	**Med**	**IQR**	**Med**	**IQR**	
height percentile		27.5	17.85–6.2	62.0	31.0–79.0	42.0	20.0–72.0	0.054
weight percentile		18.0	5.8–45.0	50.0	35.0–70.0	40.0	9.5–59.5	0.017
BMI percentile		30.0	14.5–48.8	40.0	28.0–71.0	35.0	14.5–59.5	0.204
		***n* (Total)**		***n* (Total)**		***n* (Total)**		***p*-Value**
height ≤3 perc.		2 (22)		1 (47)		1 (29)		0.342
weight ≤3 perc.		5 (22)		2 (47)		4 (29)		0.197
BMI ≤3 perc.		0 (22)		0 (47)		5 (29)		0.079

*n* = of children in a given percentile category, total = total number of children in the diet group, the *p*-value was calculated using Kruskal–Wallis test for continuous variables and Fisher’s exact test for categorical variables. *p* < 0.05 was considered as significant. med = median. perc. = percentile.

**Table 5 nutrients-14-00535-t005:** (**a**) Description of the basic characteristics of VG/VN/OM “Schoolers” (aged 4–18 years); (**b**) description of the basic characteristics of VG/VN/OM “Schoolers” (aged 4–18 years).

(**a**)								
**Group**	** *n* **	**Sex**		**Median Age**		**IQR**		***p*-Value Age**
		**Female**	**Male**					
VN	22	12	10	14.0		6.2–17.0		0.012
VG	50	27	23	7.5		5.0–9.9		
OM	30	17	13	10.2		5.6–14.4		
(**b**)								
**Variable**		**Omnivore**		**Vegetarian**		**Vegan**		***p*-Value**
		**Med**	**IQR**	**Med**	**IQR**	**Med**	**IQR**	
height percentile		55.0	36.2–74.2	42.5	19.0–59.5	53.0	33.2–74.2	0.138
weight percentile		52.5	23.5–78.8	45.0	19.0–68.8	47.5	21.2–80.0	0.524
BMI percentile		45.0	20.5–74.5	44.0	20.0–65.0	35.5	13.0–65.8	0.781
		***n* (Total)**		***n* (Total)**		***n* (Total)**		***p*-Value**
height ≤3 perc.		0 (30)		2 (22)		1 (50)		0.594
weight ≤3 perc.		0 (30)		2 (22)		2 (50)		0.250
BMI ≤3 perc.		1 (30)		0 (22)		2 (50)		0.075

*n* = of children in a given percentile category, total = total number of children in the diet group, the *p*-value was calculated using Kruskal–Wallis test for continuous variables and Fisher’s exact test for categorical variables. *p* < 0.05 was considered as significant. med = median. perc. = percentile.

**Table 6 nutrients-14-00535-t006:** B12 supplement habits in vegetarian (VG) and vegan (VN) children.

		VG (79)	VN (69)	*p*-Value
**supplement use**	yes	54	59	0.019
	no	25	10	
**regularity**	not supplementing	25	10	0.014
	irregular	2	2	
	every day	37	44	
	once a week	6	2	
	twice a week	7	3	
	three times a week	2	8	
**chemical form**	not supplementing	25	10	0.009
	methylcobalamin	23	35	
	cyanocobalamin	30	21	
	adenosylcobalamin	1	1	
	hydroxymethylcobalamin	0	2	
**medical form**	not supplementing	25	10	0.048
	drops	28	33	
	pills	26	26	

*p*-value was calculated using Fisher’s exact test, *p* < 0.05 was considered as significant.

**Table 7 nutrients-14-00535-t007:** Values of selected blood markers across VG/VN/OM groups.

	Group	Median	IQR	*p*-Value
aB12 [pmol/L]	VN	116.6	66.2–170.3	0.257
	VG	108.2	63.4–162.7	
	OM	91.4	66.3–125.1	
B12 [µg/L]	VN	545.9	410.0–789.0	0.019
	VG	572.0	397.0–849.0	
	OM	432.5	370.5–576.2	
folate [µg/L]	VN	18.1	13.7–21.4	0.057
	VG	15.9	13.5–20.3	
	OM	14.4	11.1–20.1	
Hcys [µmol/L]	VN	9.1	6.7–11.5	0.098
	VG	9.1	7.0–11.7	
	OM	9.8	8.6–12.2	
	**Group**	**Mean**	**SD**	***p*-Value**
MCV [fl]	VN	79.9	5.7	0.929
	VG	79.8	4.7	
	OM	80.2	4.4	

Cross-sectional comparison of 69 VN, 79 VG and 52 OM *p* = test (categorical variables) or ANOVA (continuous variables), *p*-value was calculated using parametric analysis of variance (MCV) or Kruskal–Wallis test. *p* < 0.05 was considered as significant. aB12 = holotranscobalamin, B12 = cyanocobalamin, Hcys = homocysteine, MCV = mean corpuscular volume.

**Table 8 nutrients-14-00535-t008:** Mean values of selected blood markers across VG/VN/OM “Infants and Toddlers” (aged 0.5–3 years).

	Group	Median	IQR	*p*-Value
aB12 [pmol/L]	VN	135.2	73.1–192.6	0.530
	VG	128.1	48.5–199.0	
	OM	107.5	72.1–165.9	
B12 [µg/L]	VN	590.0	466.0–843.5	0.278
	VG	519.0	381.0–860.0	
	OM	422.5	346.8–676.8	
folate [µg/L]	VN	18.9	16.1–22.2	0.636
	VG	18.2	15.7–20.8	
	OM	20.0	14.9–23.6	
Hcys [µmol/L]	VN	7.8	6.2–10.4	0.157
	VG	7.8	6.3–9.8	
	OM	9.0	7.4–10.5	
	**Group**	**Mean**	**SD**	***p*-Value**
MCV [fl]	VN	77.3	3.8	0.295
	VG	78.5	3.3	
	OM	77.3	3.2	

*p*-value was calculated using parametric analysis of variance (MCV) or Kruskal–Wallis test. *p* < 0.05 was considered as significant. aB12 = holotranscobalamin, B12 = cyanocobalamin, Hcys = homocysteine, MCV = mean corpuscular volume.

**Table 9 nutrients-14-00535-t009:** Mean values of selected blood markers across VG/VN/OM “Schoolers” (aged 4–18 years).

	Group	Median	IQR	*p*-Value
aB12 [pmol/L]	VN	99.2	55.9–139.3	0.420
	VG	100.8	64.0–143.6	
	OM	88.7	63.5–119.0	
B12 [µg/L]	VN	502.5	394.5–645.0	0.015
	VG	629.0	411.8–835.2	
	OM	440.0	376.5–547.0	
folate [µg/L]	VN	12.7	10.4–17.3	0.024
	VG	14.8	12.4–18.4	
	OM	11.3	9.1–15.3	
Hcys [µmol/L]	VN	10.4	9.7–14.8	0.138
	VG	9.9	7.6–12.3	
	OM	10.7	9.1–14.4	
	**Group**	**Mean**	**SD**	***p*-Value**
MCV [fl]	VN	85.4	5.1	0.001
	VG	80.6	5.2	
	OM	82.1	4.0	

*p*-value was calculated using parametric analysis of variance (MCV) or Kruskal–Wallis test. *p* < 0.05 was considered as significant. aB12 = holotranscobalamin, B12 = cyanocobalamin, Hcys = homocysteine, MCV = mean corpuscular volume.

**Table 10 nutrients-14-00535-t010:** Supplementation of vitamin B12 in mothers during pregnancy and breastfeeding.

Vitamin B12 Supplementation in Mothers	Total Number	Percent
none	3	6.5%
only during pregnancy	1	2.2%
only during breastfeeding	6	13.0%
during pregnancy and breastfeeding	36	78.3%

**Table 11 nutrients-14-00535-t011:** (**a**) Basic description of subgroup analysis VG/VN breastfed children aged 0–3 years. (**b**) Values of selected blood markers in subgroup analysis VG/VN breastfed children aged 0–3 years.

(**a**)					
**Group**	** *n* **	**Sex**		**Median Age**	**IQR**
		**Female**	**Male**		
supplementing	33	15	18	1.5	0.9–2.0
not supplementing	13	7	6	0.8	0.5–1.8
(**b**)					
**Variable**	**Supplementing**		**Not Supplementing**		***p*-Value**
	**Median**	**IQR**	**Median**	**IQR**	
aB12 [pmol/L]	157.6	96.9–224.7	37.5	33.0–42.0	<0.001
B12 [µg/L]	664.0	527.0–863.0	264.0	232.0–386.0	<0.001
folate [µg/L]	18.6	16.2–22.2	19.2	18.2–20.3	0.942
Hcys [µmol/L]	7.0	6.0–8.3	14.2	10.4–20.5	<0.001
	**Mean**	**SD**	**Mean**	**SD**	
MCV [fl]	77.3	3.5	77.8		0.665

*p*-values were calculated using parametric analysis of variance (MCV) or the Kruskal–Wallis test. *p* < 0.05 was considered as significant. aB12 = holotranscobalamin, B12 = cyanocobalamin, Hcys = homocysteine, MCV = mean corpuscular volume.

## Data Availability

Datasets used in this study are archived at the institution and can be shared by the author upon reasonable request.

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
