# Peer review of "Cross-Sectional Study of the Prevalence of Cobalamin Deficiency and Vitamin B12 Supplementation Habits among Vegetarian and Vegan Children in the Czech Republic"

_nutrients, 2022, doi:10.3390/nu14030535_

Round 1
Reviewer 1 Report
In this study the authors investigated the prevalence of cobalamin deficiency and vitamin B12 supplementation in vegetarian and vegan children. They observed high prevalence of vitamin B12 oversupplementation in children on vegetarian and vegan diets. They found no significant differences in the levels of holotranscobalamin, folate, homocysteine between groups but a significant difference in levels of cyanocobalamin. The aim of this study was interesting, however, I have the following comments and questions with regard to specific sections of the manuscript:
Abstract
- Lack of data concerning the relationship between B12 intake in vegan and vegetarian mothers during the period of pregnancy and breastfeeding and its influence on their child's growth and development. This was one of the aims of this study.
- The conclusion in this section should be modified (it is not exactly based on the results of this research).
Methods:
- Please clarify how many vegetarians and vegans was assessed: in Abstract (line 24) - ed 80 vegetarians (VG) and 71 vegans (VN), in Methods – 80 vegans (VN) and 71 vegetarians (VG) (line 100), in Results - Table 3 – 71 VN and 80 VG….
2. The groups of participants should be described more precise. The authors examined children aged 0-18 years. It would be helpful to report age as median and IQR. It is necessary to divide the examined children into age-subgroups.
- Were the omnivorous children supplemented with vitamin B12?
- Statistical analysis is not adequately done and described. How was the sample size estimated? All data were presented as mean values and SD. Did all data have normal distribution?
Results:
1. Analysis of biochemical markers in the subgroups of lactating mother and their children should be included.
- Serum levels of homocysteine should be interpreted.
Discussion:
This section should be re-writing. It is too laconic. Much of this section is taken up by the limitations of this study. There is no discussion with the literature, with data from other populations. There have been some papers on this topic in the last 2 years (Rashid et al, Eur J Haemat 2021, Bakaloudi et al, Clin Nutr 2021, Bentu Kalyan et al, Food Nutr Bull 2020, ect).
References:
- References should be written uniformly. Some words in article titles are capitalized and some are lower case.
- References 10, 11, 13, 16 should be checked.
Reviewer 2 Report
Thank you for the opportunity to review this manuscript about a Czech sample of vegetarian and vegan children and their nutritional status with vitamin B12.
I think this is a valuable piece of research in terms of the sample and area of interest but have some questions about the methodology used (please see specific comments below).
I also think the Results section should be much more concise. The more that some data in the text do not match those given in the tables (e.g. tab. 4 and tab. 5). This part requires careful checking and correction.
Moreover, in the Discussion section there are multiple descriptions of the results - there is little discussion of the obtained results in relation for research by other authors.
Abstract
Generally well written with some grammatical errors.
Unclear throughout why the purpose of the study is not the same as the purpose stated on page 2 (lines 95-97).
I suggest the Authors to consider what the main aim of the research is and maybe add the research hypothesis mentioned on page 13 (lines 424-425).
Introduction
As with abstract a few grammatical errors, but otherwise well written with good rationale.
However, I propose to supplement the first 2 paragraphs with references and add more in the remaining ones. In addition, the sentence: "Since then, there has been an increase in the popularity of plant-based diets and a change in the food market,......." needs a different reference.
I propose to consider also the connection of the sentence - lines 93 and 93 with the aim (s) of this study.
Moreover, I think it would be helpful towards the end to be a bit more specific with the objectives of the study to give the reader a better understanding of what the paper is about (this relates to my earlier point).
Materials and Methods
This section should be organized and corrected.
I propose to discuss the selection of the study design and sample collection more detail - maybe it's better to present it in the diagram - (Flow chart), including the inclusion/exclusion criteria for the study, especially/ including in relation to the second aim. It is necessary to write exactly which group (n?) was included in specific statistical analyzes.
What is the share of children in particular age groups, especially older ones, and was it the same in each of the studied groups, i.e. VG, VN and OM?
Section 2.2.: In my opinion, the text of lines 144-153 should be moved to section 2.3 and maybe better name it Biochemical analysis.
Lines 181-182: "Gandy, J., Madden, A., & Holdsworth, M. (2020). Oxford Handbook of Nutrition and Dietetics. Oxford University Press" - is not in the references.
Lines 184-185: it requires clarification whether the measurements were made with scales with a stadiometer?
The statistical tests used also require explanation. In the Data analysis section it is stated that "Analysis of variance or Kruskal-Wallis tests were carried to test for differences in blood biomarkers among vegans, vegetarians, and omnivores" while the ANOVA test is given under the tables and figures.
In addition, whether was checked before statistical analysis, the normality of variable distribution e.g with a Kolmogorov–Smirnov test?
Grammatical errors in this section are more substantial and I recommend a professional proof reader.
Results
The results should be improved.
Lines 209-216: This part should be in the chapter Materials and Methods - this is a description of the group selection.
Line 203: in total 203 children ... - this is probably a mistake - there should be 200 children (n=79 VG, n=69 VN, n=52 OM).
Lines 218-221: Sentence "Regarding supplement use, 11 parents of VN reported that they did not provide any cobalamin supplements compared with 26 VG children, while 58 parents of vegan children reported regular use of cobalamin supplements compared with 52 vegetarian children" is unclear. In addition, this applies to table 4 and should be in section 3.2.
Table 3 should present the percentage of child percentages, especially by age group, and should be broken down into 2 separate tables.
Table 7, on the other hand, can only be described in the text.
I also propose adding the sample size (n = ...) in each table and for each study group.
This section requires careful checking and correction because some data in the text do not match those given in the tables (e.g. table 4 - lines 231-231; table 5 - lines 256-257).
This chapter also requires careful grammar and language checking.
Discussion
The discussion, in my opinion, should be a more strengthened and take into account more publications in this field - concerning the population (children) different countries.
In this section there are a lots of descriptions of the results (lines 325-357) - there is little discussion of the obtained results in relation for other researches.
Major findings and conclusions should be harmonized and significantly improved, point (6) is false - "lower BMI among VN" is not true - higher percentage / number (n) of children with low BMI (≤ 3 percentile) observed. Similarly lines 223-224.
Moreover, as above, some attention to the language could improve clarity of this section.
References
More references should be added and adapted to the editorial requirements (e.g. 11. - no journal name).
Round 2
Reviewer 1 Report
The authors have revised this paper according to my suggestions.
I think that the sentence "Further...." should be removed from the aim of the study (line 106-109).
I also suggest to unify the tables and check the manuscript for errors and spaces.
Reviewer 2 Report
Dear Authors,
Thank you for considering my comments and the other reviewer. This manuscript is significantly improved and I can now support it for further processing.
However, I propose a few minor corrections before publishing, e.g.
lines 107-109: This sentence does not fit here - I suggest you delete / move it higher or edit it.
Description of Figure 1 - lines 154-162 should be placed above in the text - here rather there should be explanations of abbreviations and an explanation
the symbol should be used instead of <= 3 - ≤
Figure 2 .: Age distribution of children in dietary groups - I suggest moving to supplementary
The coding of statistical significance: <0.0001 = ****, 328
0.0001–0.001 = ***, 0.001–0.01 = **, 0.01–0.05 = *, 0.05–1 = ns (non-significant) - must be modified, eg ns (non-significant) - p <0.05; * - p <0.05; ** - p <0.01 etc.
Best regards
